# A Review of the Impacts of Roads on Wildlife in Semi-Arid Regions

**W. Richard J. Dean [1,2,\*], Colleen L. Seymour [1,3] , Grant S. Joseph [1,4] and Stefan H. Foord [5]**

[1] Percy FitzPatrick Institute of African Ornithology, DST/NRF Centre of Excellence, Department of Biological Sciences, University of Cape Town, Rondebosch 7701, South Africa; C.Seymour@sanbi.org.za (C.L.S.); karoogrant@gmail.com (G.S.J.)

[2] Wolwekraal Conservation and Research Organisation, Prince Albert 6930, South Africa

[3] South African National Biodiversity Institute, Kirstenbosch Research Centre, Private Bag X7, Claremont 7735, South Africa

[4] SARChI-Chair on Biodiversity Value and Change, Department of Zoology, School of Mathematical and Natural Science, University of Venda, Private Bag X5050, Thohoyandou 0950, South Africa

[5] Department of Zoology and Centre for Invasion Biology, School of Mathematical and Natural Science, University of Venda, Private Bag X5050, Thohoyandou 0950, South Africa; Stefan.Foord@univen.ac.za

\* Correspondence: wrjdean01@gmail.com

**Abstract:** Roads now penetrate even the most remote parts of much of the world, but the majority of research on the effects of roads on biota has been in less remote temperate environments. The impacts of roads in semi-arid and arid areas may differ from these results in a number of ways. Here, we review the research on the impacts of roads on biodiversity patterns and ecological and evolutionary processes in semi-arid regions. The most obvious effect of roads is mortality or injury through collision. A diversity of scavengers are killed whilst feeding on roadkill, a source of easily accessed food. Noise pollution from roads and traffic interferes with vocal communication by animals, and birds and frogs living along noisy roads compensate for traffic noise by increasing the amplitude or pitch of their calls. Artificial light along roads impacts certain species' ability to navigate, as well as attracting invertebrates. Animals are in turn attracted to invertebrates at streetlights, and vulnerable to becoming roadkill themselves. Genetics research across taxa confirms a loss of genetic diversity in small populations isolated by roads, but the long-term impact on the fitness of affected populations through a reduction in genetic diversity is not yet clear. Roads may rapidly cause genetic effects, raising conservation concerns about rare and threatened species. We assess mitigation measures and collate methods to identify the impact of roads on wildlife populations and their associated ecosystems, with a particular focus on recent advances.

**Keywords:** semi-arid regions; roadkill; barriers; genetic diversity

## 1. Introduction

"Long ago, in the quiet of the world, when there was less noise and more green" [1], there were also fewer roads, less traffic and fewer impacts on wildlife. To meet the needs of a burgeoning human population and rapidly enlarging economies, increasingly extensive road networks now penetrate into even the most remote regions [2], covering a considerable proportion of the land area in some countries [3,4]. Currently roads cover over 21 million km, and this overall length is anticipated to increase by 25% by 2050 [5]. Many of the areas in the world that are roadless are highly fragmented, with more than half the area in patches smaller than 1 km$^2$ [6]. Regions with more than 60% area within 382 m of a road have a high risk of ecological impacts from roads [4]. Landscape impacts and the influence of roads in a landscape context must be taken into account—"roads cannot be considered

simply as transportation lines where the influential zone is a narrow corridor; roads relate to the landscape as a whole" [7] (p 254).

Arid regions are mostly relatively remote areas that are now criss-crossed by roads [6]. Roads shape local ecosystems, especially tarred (sealed) roads in arid and semi-arid systems where runoff water increases growth along the edges of roads. Impacts to wildlife tend to be deleterious, through the greener edges of the roads and the attraction these have for animals, although some species do benefit [8–11] from this green edge [12]. Currently, arid and semi-arid areas comprise 41% of the Earth's surface area [13], and within the context of global change, drylands are anticipated to expand and cover half of the global land surface by 2100 [14]. Despite this, relatively little research has focussed on the effects of roads on the biota of arid areas. For example, in a review on the effects of roads on animal abundance, only 20% of studies had been conducted in semi-arid or arid areas [15]. Yet, the effects of roads on wildlife in arid areas might be quite different to those in environments with more complex habitat structure and where there is more species diversity, at least in birds [16]. Wildlife in mesic environments should occur at higher densities, but may be less impacted by direct collision if they avoid the open habitat presented by roads, as seen for some forest-adapted species [17]. The contrast between the road verge and the surrounding matrix differs considerably between arid and mesic regions. In forests and mesic woodlands the verges are open and the surrounding matrix is closed and complex. In arid regions, the verges are complex and often more species-rich and the surrounding matrix is open or non-existent [8]; the response of taxa (whether they avoid or are attracted) to either open or more complex habitats can have important implications for the risk of getting killed. In arid environments animals could be attracted to these islands of increased production, while animals in mesic habitats may be largely generalist species associated with clearings, edges and disturbed habitats.

The effect of noise or light pollution may be exerted over greater distances in arid environments than in those with more habitat structure, as the vegetation may help attenuate these effects. Runoff from roads in arid areas may make road verges magnets for biota as they represent areas of far higher productivity than the surrounds [12,18,19]. Road verges disturbed by clearing and mowing may facilitate invasion by non-native plants, for example in the southern African Karoo [20], but may also support rare and threatened plants and animals where the vegetation on the road verge is less degraded than the adjacent rangeland [21].

Common to both mesic and arid areas is the fact that roads themselves are associated with mortality and injury through vehicle collisions. Roads prevent wildlife access to productive places for foraging and nesting, facilitate human access to remote areas for hunting and gathering, and fragment habitats with disruptions of ecosystem processes [22,23]. Arid environments are usually open, with high visibility for vehicle users. The combination of relatively little vegetation and visibility makes the likelihood of off-road driving more likely. For example, impacts on soils (compaction) and plant communities (damage and destruction of individual plants) of ad hoc roads created by off-road recreational vehicles have been documented in the Mojave Desert [24], while in Mongolia road conditions may be such that travellers create alternative roads [25]. In arid areas, low rainfall makes for slower ecological processes, leading to lengthy return times for plants to colonise the damaged areas. Overall, roads both benefit and negatively impact a broad range of animals and plants [9,20,21,26–28]. However, the consensus on the sum of impacts to regional ecosystems is that, on balance, roads deleteriously affect ecosystems, highlighted by research on the landscape-scale impacts of road construction [8,29].

In this paper we review the effects of roads on wildlife in arid and semi-arid areas, focussing on impacts on biodiversity patterns and ecological and evolutionary processes. We also assess mitigation measures and collate methods to identify the impact of roads on wildlife populations and their associated ecosystems, with a particular focus on recent advances. For a variety of reasons, not the least primary production and nutrient cycling [30] with concomitant effects on animal and plant communities [31], road impacts on the biota in the drylands are likely to be different to the impacts in

more temperate, higher rainfall places. However, we have been unable to find any publication that explicitly addresses this question and the comparison.

## 2. Impact of Roads on Ecological Patterns

Roads impact species' abundance and distribution through both direct and indirect (secondary) effects. A wide range of taxa affected by roads is listed by Fahrig and Rytwinski [15], who categorised the species impacted by roads into (a) those species that are attracted to roads, but unable to avoid traffic (leading to roadkill), (b) species with a large home ranges, low densities and low reproductive rates, for whom roads are a barrier, (c) small animals that, for various reasons, avoid habitat near roads due to traffic disturbance or do not avoid roads but are behaviourally not able to avoid traffic, leading, again, to roadkill [15]. Almost all species listed experienced negative or neutral impacts, and all were species found in mesic habitats. The few instances of positive responses to roads included species of both mesic and dryland habitats, suggesting that they had traits or phylogenetic plasticity that allowed them to utilise roads and verges, or road verges were the only remnants of habitat in the area. Thus, it is possible that as one moves along an aridity gradient, a combination of factors associated with both biota and the roads themselves may mitigate the negative consequences of roads for biodiversity. We discuss each of these in turn.

### 2.1. Direct Alteration of Species Abundance and Distribution

The most obvious impact of roads on wildlife is mortality or injury through collision. Large numbers of animals are killed on roads [8,9], but with the exception of mammals >100 g, birds and larger reptiles, most small road-killed animals, including invertebrates, frogs and small lizards, are not recorded in any database. For reptiles, most studies have taken place in mesic areas, with relatively few reports on the rates of roadkill in semi-arid and arid areas. The numbers and species of reptiles found as roadkill likely underestimates the actual numbers killed, because their relatively small size means they are more likely to be removed by scavengers or flattened beyond recognition, leaving no evidence of collisions. In mesic northern Argentina, Cuyckens et al. [32] recorded snakes as roadkill, including a boa and a rattlesnake, but were also unable to identify 13 species of reptile that they found dead on the road. Snakes are predisposed to becoming roadkill owing to their tendency to use road surfaces for thermoregulation [33], their slow movement and need to migrate between habitat patches; other life history traits make their populations particularly vulnerable, including their long lifespans and low reproductive rates, as reviewed by Jochimsen et al. [34]. Where surveys on reptile mortalities on roads have been carried out, the rates have been high; in Idaho, where a snake was encountered every 40 km, 93% were found dead on the road [35]. Although the chances are greater of encountering a dead snake rather than a live snake along a road, these remain alarming statistics. In Australia, of 375 snakes encountered along roads in the arid MacDonnell Ranges bioregion, 34 (9%) were roadkill [36]; this relatively low incidence is possibly attributable to the nocturnal nature of many snake species in the area, when there is less traffic. A total of 164 live and 264 dead snakes were recorded along 15,525 km of road in four years in the Sonoran Desert, with the conclusion that the level of mortality could substantially damage snake populations [37]. A similar conclusion was reached by Jones et al. [38] in their studies of rattlesnakes in the Sonoran Desert, which noted that increasing road traffic contributes significantly to snake roadkill because several species move between habitat patches, crossing roads on the way. Similarly, a high incidence of road-killed snakes in the semi-arid Snake River Plain, Idaho, was reported by Jochimsen et al. [35], again with the suggestion that roads were a potential conservation threat to regional populations. Tortoise mortalities on roads and impacts due to roads in most semi-arid regions are unknown, but in the Mojave Desert, tortoise densities near roads are far lower than away from roads, with a zone of influence of up to 400 m, an effect attributed to road mortality [39].

Many bird species are killed on the roads [32,40–45] and, again, mortality may be underestimated because many are small in size. Unlike snakes in Australia [36], it seems that nocturnal birds are more

at risk [46], possibly because birds forage from the roads, which allow clear views of invertebrates in the dusk sky above (e.g. nightjars, *Caprimulgus* species [47]), and are perhaps blinded by vehicle headlights (e.g., Spotted Eagle-Owl, *Bubo africanus* [48]). In arid zones with little habitat structure, we might expect the incidence of nightjars preferentially foraging on roads at night, and consequently being killed on roads, to be lower. In southern Africa, Spotted Eagle-Owls are commonly killed on the roads [49]. In a small sample of 32 road-killed birds in the arid Kalahari savanna of South Africa, 10 alone were of this species [48]. In semi-arid southern Portugal, owls of several species are frequently killed on the roads [50]. Reasons for collisions include features of roads that influence seasonal patterns of occurrence along roads, with roads being more attractive as foraging sites in the winter, and age differences with sub-adults in at least two species, Barn Owl (*Tyto alba*) and Tawny Owl (*Strix aluco*), being more at risk [50].

Invertebrate mortalities on roads are poorly known. Many are killed, but few are identified, even to family and genus. In an area receiving ca. 500 mm of precipitation annually, roadkill of a flightless dung beetle *Circellum bacchus* (Coleoptera, Scarabeidae) in the Eastern Cape Province, South Africa, were high, with an estimated 45,000 beetles killed on the roads annually along about 58 km of road [51]. In semi-arid and arid regions, the cyclical nature of certain invertebrate abundance with outbreaks can have knock-on effects for other species [52]. For example, some species of armoured ground crickets (also known as "corn crickets," Orthoptera: Bradyporidae), including *Acanthoplus armiventris* and *A. discoidalis*, have large oscillations in population size [53], and at times of peak abundance many are killed while crossing roads (WRJD, CLS pers. obs.). All species in the family Bradyporidae eat road-killed kin [53], and these in turn are often killed while feasting. The concentration of corn crickets and brown locusts (*Locustana pardalina*) on roads also attracts other taxa, like raptors, in turn leading to increased collision rates of raptors with vehicles [54]. In Argentina, avian scavengers, including cathartid vultures, Crested Caracara (*Polyborus plancus*) and unidentified raptors were among the roadkill recorded by Cuyckens et al. [32]; similarly, three species of raptor and three species of crow were among the road-killed animals listed for the Karoo [42]; however, many avian scavengers learn to be aware of traffic and fly off on vehicle approach, thereby escaping collision.

A variety of other species may die whilst consuming roadkill. In semi-arid southern Africa, mesomammal scavengers and a diversity of small scavengers are killed whilst foraging on the road [42], which may be a general pattern for small and mesomammal scavenger species being killed on roads in semi-arid regions throughout the world. Opossums (*Didelphis marsupialis*) were the most frequent roadkill in Venezuela [43], and Pampas (Grey) Fox *Lycalopex gymnocercus* was the most frequent roadkill in Argentina, followed by Geoffroy's Cat *Leopardus geoffroyi* and another opossum (*Didelphis albiventris*) [32], suggesting that they were scavenging roadkill. All these species are about 5 kg in mass; the opossums are perhaps smaller, but all are similar in size to several species of scavenging mongooses in the semi-arid southern African Karoo. Marsupials in Australia that scavenge roadkill are small, ranging from Ringtail Possum (*Pseudocheirus peregrinus*, 550–1100 g) and Common Brushtail Possum (*Trichosurus vulpecula*, 1.2–4.5 kg) [55] to the Tasmanian Devil (*Sarcophilus laniarius*, >8 kg) [56]. In the South African arid savanna, Bullock et al. [48] found that most predator/scavenger species that were killed on the road were <5 kg, and both nocturnal and diurnal species were represented.

In addition to collisions, another direct effect of roads on wildlife populations occurs through fragmentation, which has been reviewed for tropical and temperate habitats by [10,15,57,58]; no such review has been done for arid areas [59]. The disruption of animal movement across roads appears to be equally important to fragmenting communities as a loss of genetic diversity, and many species in mesic habitats have behavioural barriers that prevent them crossing roads [57,60].

Road verges present another hazardous aspect, particularly in arid regions. Many vertebrate species find road verge habitat attractive. Most mammals and some birds that are killed are those that are residents of or frequent visitors to road verges [42]. In arid areas, the road verge represents a particularly productive environment that differs markedly from the surroundings, owing to the increased run-off from paved roads, nitrogen associated with exhaust fumes and spillage in the form of

grain and other foodstuffs. The attraction of this relatively greener verge is particularly important for herbivores, an effect that can be magnified in drought years. Both Lee et al. [61] and Klöcker et al. [62] found that kangaroos are killed on the roads more frequently in drought years, suggesting that kangaroos were attracted to the greener edges of the roads in dry times. Time of year also influences road use by other herbivores; Cuyckens et al. [32] found that roadkill in northern Argentina was more frequent in the dry season, presumably because of greener pastures along roads. The length of time spent grazing by herbivores on roadsides can also influence the risk of being killed [62], and the time of day as well as the trophic level could be important. In South African arid savanna, 78 of 124 road-killed mammals were crepuscular, 48 were crepuscular and nocturnal insectivores, 35 were diurnal herbivores, and 40 were scavenger/predators, 17 of which were crepuscular [48].

*2.2. Secondary Impacts on Species Abundance and Distribution*

Birds, small and large mammals, and amphibians appear to be markedly affected by fragmentation of rain forest habitat caused by roads [17,63,64]; the open spaces associated with roads can represent barriers to movement, but also allow predators or alien species to invade forests [64]. Similar trends have been noted in temperate forest, where roads effectively fragment the habitat of the endemic Mount Graham Red Squirrel (*Tamiasciurus hudsonicus grahamensis*) but increased habitat suitability for introduced, edge-tolerant Abert's Squirrels (*Sciurus aberti*) [65]. We know little of how arid species, already within habitats of relatively sparse vegetation structure, may respond to the open spaces associated with roads, however. It is less likely that roads would fragment habitats and local distributions of animals in the drylands, or disrupt movements across the landscape to the same extent as in mesic systems [12], and there is little information on how roads may act as barriers to animals in semi-arid and arid areas [59]. However, some chelonians and reptiles in semi-arid areas undergo population fragmentation by roads, and reductions in home ranges can be caused by roads [38,39,66]. The Mojave Desert Tortoise (*Gopherus agassizi*) had reduced home ranges near roads and fences, and the tortoises crossed roads significantly less frequently than expected by chance, indicating that the road did have some effect [39]. In contrast, four species of snake (Sidewinder (*Crotalus cerastes*), Western Diamondback (*C. atrox*), Mojave Rattlesnake (*C. scutulatus*), and the Long-nosed Snake (*Rhinocheilus lecontei*)) in the Northern Sonoran Desert showed similar distributions in habitats at two different sites adjacent to roads [38], suggesting that the partitioning effects of the road had little impact on the local snake community; the same group of species was numerically dominant in communities at both sites near roads in the Sonoran Desert.

In one of the few studies of small mammals and roads in desert areas, Garland and Bradley [67] found no relationship between home range size or life span of a number of rodent species in the Mojave Desert. All were either more abundant near the road, or, in one species, the Cactus Mouse (*Peromyscus eremicus*), less abundant near the road. This was attributed to habitat heterogeneity, rather than an effect of the road. A rodent community sampled in high-altitude desert in Utah showed no clear abundance, density or diversity effects relative to a road [19], and the conclusion was that the rodent community probably benefitted from the favourable microhabitat generated by the road. No effects of two-lane roads on demography and ecology was found for a small population of threatened San Joaquin Kit Foxes (*Vulpes macrotis mutica*) in California, with the conclusion that mitigation strategies—crossing structures and fencing—would not benefit the foxes [68].

In semi-arid and arid areas, nomadic species, and species wandering at a landscape level, use the patchiness of the area, moving to more productive areas when these are available [12,69,70]. Nomadic birds, using resources scattered in time and space, include road verges if the habitat is temporally available [70]. Population explosions of small mammals occur in Australia [71], southern Africa [72] and the South American drylands [73–75]. These suddenly abundant communities, competing for resources and space, are also more likely to utilise whatever benefits the road and road verge may bring, rather than being impacted by roads.

Corridor effects impact the movements of certain invertebrates. Two flightless carabid beetles, *Pterostichus lepidus* and *Cymindis macularis*, and an almost flightless species, *Harpalus servus*, dispersed along road verges, and the rate of movement was low compared to the rate in open areas, but no barrier effect was evident [76]. Mobile species can also be impacted by roads; carrion beetle assemblages in forests did not differ in species density, diversity and abundance across road types, but were significantly less diverse in forests near gravel roads [77].

These findings are all from mesic temperate areas; there are few studies that address the behaviour of invertebrates on roads and road verges or barrier effects in semi-arid and arid areas. Some invertebrate species may benefit from roads. Lightfoot and Whitford [78] found an association between canopy invertebrates and creosote bush (*Larrea tridentata*) on roadsides in the Chihuahuan Desert in New Mexico. In the semi-arid South African Karoo, ground-foraging ants (Hymenoptera: Formicidae) were equally diverse in species numbers in the road verge and adjacent rangeland, but road verges had relatively more rare species [79], possibly because grazing disturbance was far lower in the road verge than in the rangeland. For invertebrate communities on road sides in a semi-arid area in Israel, species-rich groups (plants, beetles, and spiders) on the road verges had higher diversity, but lacked rarity and endemism compared to communities away from roads [80]. However, species-poor scorpions and small mammals had lower diversity but were more abundant on the edges of roads [80].

Secondary effects of roads are not confined to habitat fragmentation. Exhaust fumes and dust produce environmental pollution, to which invertebrates can be particularly prone [81]. Most work to date on the effects of road-associated noise pollution on wildlife has focussed on birds and larger mammal species, and usually on the effects on density and bird song [15]. Few studies have separated the effects of noise from other road-associated disturbances [82,83]. A recent study used an array of speakers in roadless vegetation and found that road noise alone is sufficient to see a one-quarter decline in bird abundance and almost complete aversion during peak noise [84]. Given that arid areas have far less vegetation structure that might attenuate noise, the effects of noise and the distance over which these effects are felt may be considerable, although this is still a developing field of research.

Many roads are lined with street lights, and street lights are often the most concentrated and persistent sources of artificial light at night in urban areas [85]. Bats [86] and birds [87] make use of starlight and moonlight to aid navigation, so street lights may negatively impact species' ability to navigate. Bat activity seemed to be low near roads but increased three-fold at a distance of 1.6 km from the road in one study [88]. Light attracts invertebrates, so they likely suffer increased mortality owing to collisions with vehicles; in some areas, birds, some reptiles and mammals are attracted to the concentration of invertebrates at streetlights, increasing predation pressure on invertebrates and becoming vulnerable to becoming roadkill themselves. In addition, light itself may present a barrier to animal movement [89,90]. Changes to the kinds of lights used, their intensity and the time periods over which they are used could all mitigate the effects of street lighting on biota [91], and knowledge of how species respond to light could be used to help mitigate the effects of roads on animals at night [85].

## 3. Impact of Roads on Ecological Function

### 3.1. Altered Primary Production

Roads are corridors in semi-arid and desert ecosystems with resources that include runoff water from road surfaces, spillages of grain and food discarded by travellers, fresh and decomposing roadkill, dust and inputs of carbon dioxide and nitrogen from motor vehicles creating more productive (although also more polluted) patches than the surrounding habitat [8–10,12]. Total dissolved nitrogen from roads is highest near the road, decreasing with distance from the road, and is likely to have an effect on the landscape [92]. Increased nitrogen along roads is likely to benefit green growth, leading to more productive patches along roads that would draw animals from the surrounding landscape, affecting local distributions and abundance [12]. In semi-arid ecosystems in South Africa, South

America and Australia, road verges are often the only place in the landscape where there is some green growth [12,21,32,61,62] or remaining original habitat [8]. These patches can be crucial for a number of small mammal (<5 kg) [19,93,94], bird [45,94], reptile ([95] and invertebrate communities [80,81]. In some cases, the higher productivity of road verges may attract species; in other cases, road verges may be the last vestiges of available habitat [8,94,96]. Roads are known to facilitate the spread of alien invasive species, and to ease range expansions of indigenous species, usually generalists, into adjacent areas [97,98].

### 3.2. Predation and Scavenging

Animals killed on the roads, provided they are freshly killed, provide easily accessed nutrition for scavengers and predators [32,42,98–105]. Feeding on roadkill requires minimum energy, although competitive interactions between scavengers [100,105] may be costly. No energy is expended in capturing prey, but there is energy expenditure in searching for road-killed animals, and there are some risks associated with feeding on items in the road. Roadkill that is less than fresh may also offer lowered nutrient content, and for the consumer there is an unknown risk of disease from toxic organisms, including fly larvae, microbes and fungi [106].

Birds feeding on roadkill tend to be raptors and corvids, commonly observed along roads in semi-arid and arid regions [42,107–114]. Using data from 49 studies involving 234 species of mammals and birds, Benitez-Lopez et al. [19] found that raptors were more abundant near roads, with other bird taxa occurring at lower densities. The attraction for raptors, Accipitridae and Falconidae (and Cathartidae in the Americas), is two-fold; transmission poles along roads offer high vantage points from which to hunt or perch, and animals killed on the road offer food and other resources such as hair and fur for nest lining [42,115]. A third reason, applying mainly, but not only, to crows, is that transmission poles offer nest sites in areas where there are few trees. Crows build nests on crossbars on top of transmission poles in semi-arid southern Africa [116], and in similar places in the northern Chihuahuan Desert [117]. Abandoned crow nests are used by Acciptridae and Falconidae in southern Africa [49]. Most nests on poles in the Chihuahuan Desert were built by Chihuahuan Ravens (*Corvus crytoleucus*) and Swainson's Hawks (*Buteo swainsoni*), but a number of other smaller species, including passerines, also nested on poles [117].

Roadside bird survey counts in semi-arid areas are dominated by raptors and corvids. All counts in semi-arid areas exhibit some general similarities in roadside raptors [42,100,107–114,118,119]. Raptor species that occur on roadsides usually include a perch hunter or hover-hunter (predators and insectivores; e.g., small kites and kestrels), a scavenger (including omnivores, often represented by several species differing in size; e.g., large kites, vultures, and smaller carrion-eating "hawks"), a coursing species (predators, represented by harriers), and in road verges with taller trees, a predating bird hunter such as a sparrowhawk or falcon. Categories of species (predators, scavengers, insectivores, omnivores) in roadside raptor counts in semi-arid South America and semi-arid southern Africa differ markedly in all diet categories (Table 1).

**Table 1.** Road count data shown as percentages of trophic categories; data extracted from [103,109] and unpublished data collected by W.R.J. Dean. Pied Crows and White-necked Ravens have been included in this table as 50% omnivores, 50% scavengers.

| Trophic Level | Southern Africa (*n* = 13,700) | South America (*n* = 934) | Australia |
|:---:|:---:|:---:|:---:|
| Predator | 31.9 | 11.7 | 46.0 * |
| Scavenger | 19.9 | 15.2 | |
| Insectivore | 29.9 | 54.5 | 13.3 |
| Omnivore | 18.2 | 18.5 | 40.0 |

* Some species categorised as predators in Australia also function as scavengers, including Whistling Kite (*Haliastur sphenurus*), Wedge-tailed Eagle (*Aquila audax*) and Black Kite (*Milvus migrans*) [103].

Some similarities between the avian scavengers in southern Africa and those in the deserts of the southwestern USA are apparent. Avian scavengers are dominated by cathartid vultures in Mexico [113], Venezuela ([43], Peru [109], for a number of other semi-arid sites, including Brazil, southern Argentina, Chile and Peru [109,119] and northern Argentina, judging by the number killed on the roads [32]. However, in North American deserts, cathartid vultures are less frequent, and the avian scavengers are Common Ravens (*Corvus corax*). Carrion from road-killed wildlife was thought to be a major factor in drawing ravens to roads in the Mojave Desert [110] and in the Karoo, southern Africa [99]. The avian scavenger component in Australia lacks cathartid and Old World vultures, and is made up of several predators and omnivores, of which only the Australian Raven (*Corvus coronoides*) seems important [55,103], and possibly the Torresian Crow (*Corvus orru*) since this species was recorded as roadkill [120]. In semi-arid southern Africa the avian scavenger group is dominated by three species of corvid, and not by vultures [42,98–100,114]. The availability of roadkill changes the local distributions of predators and scavengers [110,118,121].

Scavenging on roads has two other ecological effects: (1) increased incidence of death of scavenging species themselves, and (2) apparent competition, an indirect effect that occurs when one species has a negative effect on the abundance of another species mediated through the action of shared natural enemies. In this case, carrion has a negative effect on prey species in the vicinity of roads, and the carrion acts to attract scavenger-predators, which in turn prey on biota in the vicinity of the road [98].

## 4. Impact of Roads on Evolutionary Processes

### 4.1. Genetic Effects

A large and emerging body of genetics research across taxa confirms a loss of genetic diversity in small populations isolated by roads [122–125]. However, long-term impacts on the fitness of affected populations through the reduction in genetic diversity is not yet clear [123]. Roads often decrease functional connectivity through barriers and increase the genetic distance among individuals in populations, although such barriers are often not complete [123,125]. However, in Desert Bighorn Sheep (*Ovis canadensis nelson*), roads blocked gene flow and caused a rapid decline in genetic diversity in as little as 40 years [126]. In agricultural areas, where there is minimal habitat along roads for some species, roads may form a complete barrier, cutting off gene flow, with a significant isolation effect [127]. Flightless invertebrate species are particularly vulnerable to roadkill, and even if the species are averse to crossing roads, barrier effects, preventing movements by this group, can lead to inbreeding depression and pose a threat to population viability [124,127,128]. Roads may thus rapidly cause genetic effects, raising conservation concerns about rare and threatened species [123].

### 4.2. Roads Exerting Selective Pressure on Populations

A study conducted in Poland found that, unlike predators, which remove individuals that are weak or in poor condition, roads "predate" birds randomly [129]. Thus, "fitter" birds are removed through roadkill, whereas predators consistently remove weaker individuals from the population [130]. However, the evidence for the removal of weaker individuals is controversial and inconsistent, and may be species-specific according to predator and prey [131].

In semi-arid areas, environmental variability leads to "boom or bust" [132], with populations of potential prey being abundant or scarce. During droughts, individuals may be weakened by shortages of food and greater energy expenditure on foraging. Reducing exposure to attack by predator species increases the chances of survival, but starvation risk lowers vigilance, decreasing survival [133]. Roads in semi-arid areas are likely to attract animals in dry times, or individuals may range further in search of resources, crossing or associating more frequently with roads at these times, with both 'strong' and 'weak' individuals at risk from roadkill and predators.

### 4.3. Evolution of Behaviour to Counteract Road Noise

Traffic noise impacts certain species, including invertebrates [134], anurans [135–137], and birds [84,138]. Noise pollution from roads and traffic interferes with vocal communication by animals. By implication, birds living along noisy roads would compensate for the traffic noise by increasing the amplitude or pitch of their calls. Birds in cities sing at a higher frequency than conspecifics in rural areas [139], and males at noisier locations sing louder than birds in territories less affected by background sounds [140]. We might expect to see adaptations to noise in calls in arid areas, but their relative importance and how habitat structure, or the lack thereof, might interact with these changes is not known. As mentioned earlier, traffic noise is associated with a marked decline in bird abundance, even in an experimental roadless area [84].

There have been several studies on evolutionary adaptations in animals living on roadsides in the mesic regions, but none have been documented for drylands. Developmental plasticity in grasshoppers led them to adjust their signals to counteract road noise [134]. Grasshopper males in roadside habitats produced songs with higher frequency components compared to songs of males from non-roadside habitats [134], possibly by selection for minimizing signal masking by road noise. As in grasshoppers, frog calls are innate, not learned, so shifts in calling frequency under noisy (road traffic) conditions may represent an evolutionary adaptation. A comparison of the calls of two frog species, the Southern Brown Tree- frog (*Litoria ewingii*) and Common Eastern Froglet (*Crinia signifera*), living in ponds, lakes, dams and quiet pools within streams in southern Australia suggested that there has been an evolutionary shift in calling to compensate for traffic noise [136]. The Southern Brown Tree Frog calls at a higher pitch in traffic noise and it is suggested that this shift in call frequency is enough to benefit the caller, but calling at a higher pitch where there is road noise is less than certain in the Common Eastern Froglet. An experimental test of traffic noise impacts on anurans, similar to the test carried out on birds by McClure et al. [84], was done by Grace and Noss [141]. The authors created a 'phantom road' by playing different traffic noise treatments in three roadless areas, and concluded that avoidance of traffic noise could increase the success of communication with conspecifics.

### 4.4. Evolution of Morphology to Avoid Being Killed on the Road

The evolution of morphological adaptations in birds living on roadsides is known only for one species. The majority of road-killed Cliff Swallows (*Petrochelidon pyrrhonota*) adjacent to roadside nesting sites in southwestern Nebraska were found to have relatively longer wings than the 'population at large'. Selection for shorter wings was thought to have evolved in response to mortality on the road; birds with shorter wings are more manoeuvrable and able to avoid traffic, although this may not be the only reason that wing length had changed [142]. This is another area that is ripe for research.

## 5. Monitoring Road Impacts

### 5.1. Identifying Hotspots

Given that roads cover such large areas and occur from major highways to remote roads in rural areas, citizen science can make a valuable contribution to data collection. A recent study of South Africa found that roadkill reports from citizen scientists broadly identified the same hotspots and taxonomic groups in roadkill as reports from trained observers, although there was a slight bias in citizen scientist reporting towards more charismatic and easy-to-identify species [143]. Nevertheless, these data sources can help identify areas and taxa requiring further investigation and mitigation measures [144].

### 5.2. Modelling Approaches

Recently, the use of encounter theory with decision analysis was used to identify hotspots for collisions between speedboats and manatees [145]. It is possible that such approaches could be applied

to data on roadkill and used to implement traffic calming measures that might reduce vehicle-wildlife collisions on roads.

### 5.3. Populations at Risk

Animal species populations most at risk from roads are those that are not affected by road noise or traffic, and that regularly and seasonally move across roads to reach patches of habitat, or forage along road verges. However, the risk to populations may be variable according to environmental conditions such as drought that draw animals to more available resources along roads, or outbreaks of 'plague' invertebrates, such as locusts, that cross roads in high numbers [52], attracting insectivores that are themselves killed on the road. Other factors also influence roadkill, including the types of roads and human activity, such as hunting, away from the road. Greater Kudu (*Tragelaphus strepsiceros*) in the semi-arid Eastern Cape Province, South Africa, move across roads more frequently (and to some extent unpredictably) in the rutting and hunting seasons, with concomitant hazards for vehicles and an increase in the numbers of animal-related accidents [146,147]. Kangaroos in Australia are frequently killed on the roads, predictably near curves in the road and near sources of water [61], but this may differ according to environmental conditions, with some species (Red Kangaroos (*Macropus rufus*) and Euros (*Macropus robustus erubescens*)) more frequently killed during droughts [61,62].

### 5.4. Spatial Effects

Roads that run through woodlands, forests, wetlands, and over drainage lines where animals are likely to move across the road, or the road disrupts any movement corridors [144,148], or where there is remnant habitat along the road [8,94,96] are high-risk for a large number of species in both mesic and arid environments. Season is also important; Servals (*Leptailurus serval*) were more frequently killed on the road in the dry season, and where there were adjacent wetlands in South Africa [144]. Similarly, kangaroos are killed more often on roads through open plains than other places [62]. Other factors such as environmental conditions, somewhat independent of the position of the road in the landscape, can influence frequency and abundance of roadkill (see Section 2). The width of the road may be important—invertebrates and slow-moving small vertebrates would spend more time crossing a wide road, and therefore are at higher risk than fast-moving animals.

## 6. Mitigation of Road Impacts

Most of the measures that have been proposed to mitigate the impacts of roads on animals have been aimed at the motorist or controlling animals' access to roads [149]. Over 40 types of road mitigation measures that aim to reduce roadkill are available, but these measures vary greatly in terms of cost and effectiveness [150]. Fencing has been suggested to mitigate roadkill [150]; this is supported to some extent by Clevenger et al. [151], but is not particularly effective for Greater Kudu in South Africa [146], who are able to clear a 2-m fence in one jump [152]. Long stretches of 'game-proof' fencing in the Eastern Cape, South Africa, partly clustered roadkill sites of Greater Kudu, whereas roads with few or no game fences showed a random distribution of roadkill [146]. Similarly, Clevenger et al. [151] found that roadkill was greatest within 1 km of fence ends. The initial and maintenance costs of fencing are high, but in general they may reduce roadkill of large mammals by 80–95% [150]. Fencing, as a means to effectively mitigate road mortality, should be viewed in the light of the objectives—if as a result of fencing, habitat connectivity is reduced, then it has another undesirable impact [153]. The length of the fencing is important; long fences are more effective at reducing roadkill, but there is still a fence-end issue [149]. Proposed crossing structures include underpasses (pipes, drainage culverts), bridge-like overpasses and canopy rope bridges [57]. Crossing structures with associated fencing are no more effective than fencing alone [150], but, according to Sielecki [154] and Clevenger et al. [151], can reduce ungulate-vehicle collisions by 80% or more. Underpasses and overpasses reduce the barrier effect of fencing alone and provide relatively safe crossing places for a wide range of species [149,150].

Animal detection systems, although reducing roadkill, have only a limited effect [155]. In the past, mitigation of animal-vehicle collisions was largely addressed by putting up signs to warn motorists of the potential for animals to be on the road (e.g., for Greater Kudu and tortoises in South Africa, kangaroos in Australia) but these have a limited effect [48]. The problem needs to be approached from the other direction. Animal detection and animal warning systems located in the right-of-way are cheap and may be effective in the long term. Vehicle-based animal detection systems [149,155] operate on a different principle as they warn the animal that a vehicle is approaching, which may be too late for both the vehicle and the animal concerned. All animal detection and warning systems in the world that were known at that time were reviewed by Huijser and McGowen [155]. Subsequently, Huijser et al. [149] reviewed 39 mitigation measures to reduce roadkill and reduce barriers for wildlife across roads, including roadside animal detection systems (RADS) that may be effective in controlling mortality on roads. The objective of the RADS system is to warn drivers with flashing signs that there are animals near the road. A recent study [156] suggests that RADS can be effective in controlling mortality on roads. A significant reduction in vehicle speed was detected when the RADS were activated, but this varied seasonally, with only tourists taking note of the signs and slowing down, but local residents, presumably habituated to the signs, not showing any response; nevertheless, a reduction in vehicle speed was noted and thought to be important in reducing roadkill.

An animal-based approach was used by Riginos et al. [157] in Wyoming to reduce collisions between large ungulates and vehicles. Wildlife warning reflectors, designed to reflect headlights and alert large animals to approaching vehicles, were deployed along roadsides. Results of tests yielded mixed results, but an accidental fortuitous finding was that the white canvas bags used to cover reflectors not in use was effective in stopping Mule Deer (*Odocoileus hemionus*) from crossing the road. The authors concluded that the simple white canvas on reflector poles was substantially more effective in reducing roadkill of deer, and suggest that this new vigilance-enhancing mitigation method should be explored. It has the advantage of low technology and cost.

## 7. Directions for Future Research

Disruption of pollination services through road-killed invertebrates has been investigated in mesic areas [158], but the effects of roads on invertebrate pollinators and pollination in semi-arid areas is not known. Given that roadsides in arid areas are often the foci of more productive vegetation (and therefore places where flowers would be more abundant), the impacts of roads on invertebrate pollinators need to be investigated. The species of plants that constitute the landscape-wide flowering events in Namaqualand, South Africa are all pollinated by insects [159,160] (and see [161,162] for Rhenosterveld vegetation in the same general area), but nothing is known of the effects of roads on the pollinator community. Donaldson et al. [161] examined plant reproductive success in Rhenosterveld fragments, not necessarily caused by roads, and found varied pollinator diversity and seed set according to fragment size.

Vertebrate pollinators may also be affected if they avoid roads, road noise and light. It has been shown that there is reduced pollination in sunbird-pollinated heathers (*Erica* species) along roads in the temperate Cape Floristic Region, South Africa [163]; this may be due to the birds avoiding traffic noise. Alternatively, bird species that can tolerate noise may be associated with improved pollination and seed-set (see [164] for an example from a mesic environment). Future research would need to identify the positive and negative effects of noise on the behaviour and relative densities of pollinating species in arid environments, to ascertain just how roads may impact pollination and seed set. Road noise may also impact seed-dispersing animals; alternatively, roadside vegetation may attract these species, but again, little research has been conducted on how roads affect ecological processes in arid areas.

## 8. Conclusions

For several reasons, the impacts of roads and associated road verges in semi-arid and arid areas appear to differ, quite markedly in some cases, from the impacts in temperate, higher-rainfall areas.

Soils in semi-arid and arid areas lack leaching, and are usually not nutrient-poor [30], but the growth of vegetation is limited by water. Runoff water from roads supports vegetation that may be the only habitat available for all taxa at certain times of the year, and therefore attractive, with the result that animal communities along roads may be relatively higher than those in temperate and rainforests, where animals might tend to avoid open areas associated with roads, so all taxa in arid areas are probably at more risk of becoming roadkill. Alternatively, because of the stochastic nature of the environment, animal communities probably lack stability in the drylands, and the use of roads and their verges is dynamic according to environmental conditions [70]. In addition, animals often need to roam larger areas in their search for resources in drier periods within these drylands, which exposes them to more road crossings and activity near roads, as well as increasing the chances of juvenile individuals being on roads. We can speculate that, because arid areas have far less vegetation structure that might attenuate noise, the effects of noise and the distance over which these effects are felt may be considerable, and different to the effects of noise in high-rainfall areas with denser and higher plant growth. Many animals appear to be markedly affected by the fragmentation of forest habitat caused by roads, presenting barriers to movement, but dryland animals, already within habitats of relatively sparse vegetation structure, may not all respond in the same way to the open spaces associated with roads in temperate areas. It is less likely that roads would fragment habitats and local distributions of animals in the drylands, or disrupt movements across the landscape to the same extent as in mesic systems, but there is little information on how roads may act as barriers to animals in semi-arid and arid areas. The impacts of roads on animals (and plants) in semi-deserts and deserts would be a rewarding field for future research, if only to investigate some key areas such as the importance of roadkill in local food webs, and the impact of the removal of individuals by roadkill on the structure of the local animal communities.

**Author Contributions:** Conceptualization, C.L.S., W.R.J.D., G.S.J., S.H.F.; Original draft preparation, W.R.J.D., C.L.S.; Review and editing, all authors; Funding acquisition, S.H.F.

**Funding:** This publication is funded by the DST-NRF Centre of Excellence for Invasion Biology, through the South 380 African Research Chairs Initiative Chair on Biodiversity Value and Change in the Vhembe Biosphere Reserve, hosted by the University of Venda, Private Bag X5050, Thohoyandou, 0950 South Africa.

**Conflicts of Interest:** The authors declare no conflict of interest.

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
