# Peer review of "A Review of the Impacts of Roads on Wildlife in Semi-Arid Regions"

_diversity, doi:10.3390/d11050081_

Round 1

Reviewer 1 Report

This review is comprehensive and one of the more complete ones I have seen on this topic.  That said, it doesn't provide much new insight into road mortality, other than highlighting work done in semi-arid areas. 

My one main comment is that it would be nice to better justify early in the manuscript why a review specifically for these areas is needed; what are road impacts likely to reveal in arid regions that might be different from less arid areas?  More explicit comparisons between arid and non-arid regions--why they are similar and why they are different--would be useful throughout the manuscript.

Author Response

Please find the response attached.

Reviewer 2 Report

This is a nice review. I made a number of comments directly on pdf file, but they are easy to incorporate. A half-page chapter with conclusions would be a nice end.

Author Response

Please find the response attached.
